# Estimation of the Measurement Accuracy of Wireless Passive Resonance Sensors

**DOI:** 10.3390/s25030747

**Published:** 2025-01-26

**Authors:** Leonhard M. Reindl, Taimur Aftab, Thomas Schaechtle, Thomas Ostertag, Wei Luo, Stefan Johann Rupitsch

**Affiliations:** 1Laboratory for Electrical Instrumentation and Embedded Systems, Faculty of Engineering, University of Freiburg, 79110 Freiburg, Germanystefan.rupitsch@imtek.uni-freiburg.de (S.J.R.); 2RSSI GmbH, Bürgermeister-Graf-Ring 1, 82538 Geretsried, Germany; thomas.ostertag@rssi.de; 3School of Integrated Circuits, Huazhong University of Science and Technology, Wuhan 430074, China; hustluowei@gmail.com

**Keywords:** resonator, quality factor, wireless, passive, chipless, estimation theory, statistical error, measurement accuracy, Crámer–Rao lower bound, Fisher information matrix, maximum likelihood estimation, Aboutanios algorithm

## Abstract

Resonators are passive devices that respond to an excitation signal by oscillating at their natural frequency with exponentially decreasing amplitudes. Physical, chemical and electrical variables can modify the natural frequencies of resonators. If resonators are connected to antennas or other transducers that couple into a communication channel, they enable purely passive sensors that can be read wirelessly. In this manuscript, we use maximum likelihood estimation to analyze the measurement accuracy that can be achieved by the wireless readout of passive resonant sensors as a function of the read signal, the stimulation power and noise figure of the reader, the distance and transducer gain of the transmission channel, and the natural frequency and quality factor of the resonant passive sensor. The Crámer–Rao lower bound characterizes the minimum variance of the natural frequency and decay constant of the resonator. We show the derivation of the Crámer–Rao lower bounds from the Fisher information matrix based on a maximum likelihood estimation of discrete-time samples of an exponentially decaying phasor. This theoretical lower limit of accuracy is almost achieved by an iterative algorithm that approximates the maximum of the measured resonator spectrum with a Lorentz curve.

## 1. Introduction

Sensors are essential parts of any control system. Wireless sensors, called transponders, are used at moving or rotating control points, on or in living beings, and in situations where wiring would be expensive. Passive transponders are used in applications where traditional battery, RFID or energy harvesting-based radio technology cannot be operated or only at high operating costs. Chipless passive transponders require neither a battery nor an integrated circuit. Such a transponder system consists of two parts, a reading unit and passive transponders that act as cooperative targets. Both parts are connected by transducers via a wireless connection, for example a radio, inductive, capacitive or ultrasonic connection, as shown in Figure 1. To stimulate the passive transponders, the reader sends a read signal to the transponder nodes via the communication channel.

Three types of chipless passive transponders are described in the literature: delay lines, resonators and mixer types [1]. This contribution focuses on resonators with a high quality factor *Q*, where the quantity being measured changes the resonant frequency of the resonator. The read signal stimulates an oscillation in the resonator, which decays exponentially after the read signal is switched off. A part of the energy stored in the oscillation is sent back to the reader as a backscatter signal, where it is received, sampled and evaluated. The resonant frequency can be extracted if the decay time of the oscillation is long enough to separate it in time domain from the read signal and some ambient echoes in the wireless transmission channel. Mixer types sometimes mix the read frequency down to an intermediate frequency, which then also excites a resonator [2,3].

So far, ceramic dielectric resonators [2,4], bulk [5,6] and surface acoustic wave resonators (SAWs) [1,7,8,9,10,11], HF cavity resonators [12,13], LC resonators [14,15,16], coplanar resonators [4,17], spiral resonators [18], and air-filled substrate-integrated waveguide resonators [19] are presented in the scientific literature for wireless temperature [6,11], pressure [1], blood pressure [20], torque [7,21], strain [4,10,12,19], transient load [9], vibration [22], mass flow [4], corrosion of steel [23], pH value [16], and food quality [24] measurements together with other physical parameters. The review [25] provides a good overview of the current state of research, development and applications of wireless resonance sensors and [26,27] of the subset of passive LC wireless sensors.

Due to the low complexity of chipless passive transponders, which enable operation without batteries or electronic circuits, the instrumentation technology is assessed as maintenance-free, robust and suitable for use in harsh environments. Systems using chipless passive transponders usually surpass the read distance of RFID-based transponder systems by far, since they are linear time-invariant systems where the read distance is limited only by the receiver noise, which blocks detection of the response signal from distances beyond the maximum read distance.

There are three types of error sources: systematic errors, statistical errors and unknown systematic errors [28]. This manuscript addresses the evaluation of noisy exponentially decaying complex exponential functions, where the noise is assumed to be Gaussian distributed and distortion-free. In addition, linear, time-invariant transmission channels without interference from other transmitters as well as linear and time-invariant transponders with constant natural frequencies are assumed. Due to the linear, time-invariant characteristics of the transducer and the resonator, a capacitive or inductive load on the transducer can pull the natural frequency of the resonator and thus distort the measurement result [27]. In the case of non-time-invariant transmission channels, for example if the distance between the reader and transponder varies, in the case of temporary shadowing, or if stochastic interference occurs, more complex methods, as described in [29,30], must be combined with the algorithms presented here.

Y.-X. Yao and S. M. Pandit were the first to calculate the theoretical lower bounds, so-called Crámer–Rao lower bounds, for the parameters of a sampled exponential decaying complex phasor in noise based on a maximum likelihood estimation [31]. Based on their results, this manuscript calculates a theoretical lower limit of the measurement accuracy of a wireless readout of a passive transponder system using resonant sensor nodes.

There is an increasing number of publications on extracting the resonance frequency and quality factor of a noisy complex exponential function with exponentially decreasing amplitude. Examples are a numerical fit of Lorentz curves to the measurement data [9,32], a frequency estimation based on linear prediction and singular value decomposition [33,34], or super-resolution based methods [35]. An overview and performance comparison of some concepts can be found in [27,36,37]. Using sin(x)/x interpolation, analytic solutions for some frequency estimators were presented [38]. What is still missing, however, is an analysis that allows to predict the theoretically possible measurement accuracy based solely on the physical and electrical parameters of the measuring system. By comparing this theoretical measurement accuracy with the actual one, the performance of the measurement system can be assessed qualitatively and quantitatively. In addition, it can be predicted which measurement accuracy can be expected at which distance.

E. Aboutanios presented an iterative, high-performance algorithm which approximates the points around the maximum of the measured spectrum by a Lorentz curve, thereby reaching an accuracy of the Crámer–Rao lower bounds [39,40]. Using this algorithm, several examples of the achievable accuracy are shown for wireless readout of resonators of different quality factors in typical ISM (Industrial, Scientific, Medical, ISM) [41] and SRD (Short-Range Device, SRD) [42] bands as a function of read distance or signal-to-noise ratio.

The manuscript begins by presenting a mathematical model of a noisy discrete-time complex signal with exponentially decaying amplitude. When reading a passive resonator remotely, the initial signal-to-noise and interference ratio results from the radar parameters’ transmission power, gain of the transducer to the transmission medium, reading distance and stimulation duration as well as from the transponder parameters’ resonance frequency, quality factor and impedance matching of the resonator to the transducer. The equations for the maximum likelihood estimation of the parameters of the exponentially decaying discrete-time signal and their Crámer–Rao lower bounds are then formulated. An iterative numerical algorithm for determining these parameters is also presented, and its performance is compared with the theoretical lower bounds.

All visualizations of the formulas and the numerical simulations were carried out using MATLAB, Version 9.13.0 [43].

## 2. Signal-to-Noise and Interference Ratio by a High-Q Resonator as Cooperative Target

To calculate the maximum achievable measurement accuracy of a passive transponder system using high-Q resonant sensors, we need the signal-to-noise ratio of the received signal in the reader. In this section, we first calculate the signal power of the signal backscattered by the passive transponder, which is received and sampled in the reader, using the radar equation and Friis’ law [44]. The noise power contained in the signal is then calculated as thermal noise. At the end of the section, the elimination of interference caused by waves backscattered elsewhere in the transmission channel is discussed.

### 2.1. Signal Power

Various radar techniques for readers for passive chipless transponders have been presented in the literature such as time domain sampling, frequency domain sampling and hybrids [45,46,47,48]. The highest signal-to-noise ratio and thus the highest measurement accuracy of the frequency is obtained by the reading device sending out a CW signal with a carrier frequency ω=2πf very close to the natural frequency of the resonator as the read signal and sampling the decaying resonator signal after switching off the stimulation.

The reader transmits this signal with a power $P_{t}\left(t\right)$ which is sent via a transducer, e.g., an antenna with the gain $g_{t}$ over a wireless channel. Electromagnetic or acoustic channels have been used as wireless channels for the far-field transmission of the read and the backscattered decaying signal [5,6]. The transponder receives a fraction $P_{r}\left(t\right)$ of the emitted signal at a distance R>λ, where λ is the wavelength in the transmission channel. If the transponder’s transducer gain is given by $g_{r}$, then $P_{r}$ is ideally given by the Friis equation [44](1)$$P_{r}=P_{t}\cdot g_{t}\cdot g_{r}\cdot {\left(\frac{\lambda }{4\pi R}\right)}^{2}.$$

An additional propagation loss $L_{a}$ might be caused by absorption of the wave in the media [9]. The absorption per unit distance is given by an absorption coefficient of the media $\alpha_{m}$. The received power $P_{r}$ is then given by(2)$$P_{r}=P_{t}\cdot g_{t}\cdot g_{r}\cdot e^{-\alpha_{m}R}{\left(\frac{\lambda }{4\pi R}\right)}^{2}.$$

The transmission of signals between the reader and the passive transponder does not correspond to free space propagation if the signals at the receiver are changed by superimposing signal components reflected in the transmission channel. If the quality of the resonator is significantly higher than the quality of the transmission channel and neither the mutual arrangement nor the distance between reader and transponder changes during communication, a constant effective distance can be specified with which the received power can be determined according to Equations (Equation 1) or (Equation 2). The effective distance sometimes differs from the geometric distance.

The signal $s_{load}\left(t\right)$ at the feed point of the transponder antenna can be written through(3)$$s_{load}\left(t\right)=\sqrt{P_{r}}\cdot e^{j(\omega t+\varphi_{0})},$$
where the phase shift $\varphi_{0}$ is given by the delay caused by the transmission between the reader and transponder.

$H_{R}\left(\omega \right)$ describes the response of the resonator loaded with the impedance of the transducer by a signal at the frequency ω. The total received power $P_{r}$ is absorbed by the resonator when the source resistance of the transducer is matched to the impedance of the resonator at resonance and when the carrier frequency of the read signal is close to the natural frequency of the resonator. The resonator then begins to oscillate in a forced oscillation.

Incomplete matching of the antenna to the resonator reduces the power transfer twice, during stimulation from the antenna into the resonator and during decay from the resonator to the antenna, $H_{R}\left(\omega \right)$ thereby is reduced by the mismatch loss. The damping constant α of free oscillation lengthens slightly in this case because the resonator is operated somewhere between open and loaded.

The vibration characteristics of the resonator is described by the natural angular frequency of the undamped oscillation $\omega_{0}=2\pi f_{0}$ and the loaded quality factor *Q*, which gives the damping constant α of free oscillation(4)$$\alpha =\frac{\omega_{0}}{2Q}=\frac{\pi f_{0}}{Q}.$$

The natural angular frequency of the damped oscillation $\omega_{d}=2\pi f_{d}$ is given by(5)$$\omega_{d}=\sqrt{{\omega_{0}}^{2}-\alpha^{2}}.$$

After the load time $t_{load}$, the stimulation signal is switched off, and the oscillation will decay at its damped natural angular frequency $\omega_{d}$ with the signal $s_{decay}\left(t\right)$, which will start at the power $P_{decay}$ [49](6)$$s_{decay}\left(t\right)=\sqrt{P_{decay}}\cdot e^{\left(-\alpha +j\omega_{d}\right)\left(t-t_{load}\right)+j\varphi_{1}}.$$

The accumulated phase shift $\varphi_{1}$ and the power $P_{decay}$ at the start are given by(7)$$\begin{array}{ccc} \varphi_{1} & = & \varphi_{0}+\omega t_{load} \end{array}$$(8)$$\begin{array}{ccc} P_{decay} & = & P_{r}{H_{R}}^{2}\left(\omega \right){\left(1-e^{\left(j\omega_{d}-j\omega -\alpha \right)t_{load}}\right)}^{2}. \end{array}$$

In Figure 2, the dashed lines in blue show the function of time during which the stimulation signal is provided by the antenna due to the picked-up read signal and in a solid black line the oscillation which is built up in the resonator. The amplitude of the excitation signal is normalized to one in the image. The frequency of the excitation and the decaying resonator is plotted in $f/f_{0}$ and the time in $t\cdot f_{0}$. The stimulation signal stops at $t_{load}$ and the oscillation decays. The signal of the decaying resonator is divided between the source impedance of the antenna and the internal resistance of the resonator.

Near resonance, i.e., $\omega \approx \omega_{d}$, the term $\left(1-e^{\left(j\omega_{d}-j\omega -\alpha \right)t_{load}}\right)$ increases for $\alpha t_{load}<1$ linear with $t_{load}$ and approaches 1 for $\alpha t_{load}>\pi$. The decaying oscillation in Equation (Equation 6) supplies the response signal, whereby, if matched ($H_{R}\left(\omega \right)=1$), half of the power is dissipated in the internal resistor of the resonator and half is coupled to the transducer and sent out as a backscattered wave with power $P_{backscattered}$ to the reading device:(9)$$P_{backscattered}=\frac{1}{2}\cdot P_{decay}$$

This signal again propagates through the media back to the reader. The reader, at the distance *R*, picks up a fraction $s_{in}\left(t\right)$ of the backscattered signal, which will start at the power $P_{in}$
(10)$$\begin{array}{ccc} s_{in}\left(t\right) & = & \sqrt{P_{in}}\cdot e^{\left(-\alpha +j\omega_{d}\right)\left(t-t_{load}\right)+j\varphi_{2}} \end{array}$$(11)$$\begin{array}{ccc} P_{in} & = & P_{backscattered}\cdot g_{t}\cdot g_{r}\cdot e^{-\alpha_{m}R}\cdot {\left(\frac{\lambda }{4\pi R}\right)}^{2}\\ & = & \frac{1}{2}\cdot P_{r}{H_{R}}^{2}\left(\omega \right){\left(1-e^{\left(j\omega_{d}-j\omega -\alpha \right)t_{load}}\right)}^{2}\cdot g_{t}\cdot g_{r}\cdot e^{-\alpha_{m}R}\cdot {\left(\frac{\lambda }{4\pi R}\right)}^{2}\\ & = & \frac{1}{2}\cdot P_{t}\cdot {g_{t}}^{2}\cdot {g_{r}}^{2}\cdot e^{-2\alpha_{m}R}\cdot {\left(\frac{\lambda }{4\pi R}\right)}^{4}\cdot {H_{R}}^{2}\left(\omega \right)\cdot {\left(1-e^{\left(j\omega_{d}-j\omega -\alpha \right)t_{load}}\right)}^{2}. \end{array}$$

Figure 3 shows the initial received power $P_{in}$ at the end of stimulation in the reader for ISM [41] and SRD [42] bands that are often used to read passive resonant transponders. The initial received powers are independent of the quality factor *Q* of the resonator. Free space electromagnetic propagation was assumed with $P_{t}=10\;mW$, $g_{t}=g_{r}=2$, $\alpha_{m}=0$, $H_{R}\left(\omega \right)=1$, i.e., an stimulation at resonance frequency, and $\alpha \cdot t_{load}=\pi$, i.e., an excitation via *Q* oscillations. The curves are shown from a distance between the reader and transponder of R≥λ.

The phase shift $\varphi_{2}$ in Equation (Equation 10) results from the delay caused by the transmission from the reader to the transponder due to the stimulation time $t_{load}$ and by the re-transmission from the transponder to the reader.

For electromagnetic waves in the UHF and VHF range, $\alpha_{m}\approx 0$. If the stimulation time $t_{load}$ is in the order or larger than $Q/f_{0}$, then the resonator is maximally stimulated and we can neglect the phase shift due to the interference between the forced oscillation and the natural oscillation in the resonator [49]. If further, the antenna is matched to the impedance of the resonator at resonant frequency, and the interrogation frequency is close to the natural frequency, i.e., $H_{R}\left(\omega \right)\approx 1$, then Equation (11) simplifies to(12)$$P_{in}\approx \frac{1}{2}\cdot P_{t}\cdot {g_{t}}^{2}\cdot {g_{r}}^{2}\cdot {\left(\frac{\lambda }{4\pi R}\right)}^{4}.$$

### 2.2. Interference Due to Environmental Echoes

After switching off the read signal, the reader receives the signals not only from the decaying oscillation of the resonators but also from a variety of other echoes from the environment. A wireless measurement of the natural frequency of a resonant passive transponder is only possible if the decaying signal by the resonator lasts significantly longer than all ambient echoes picked up by the reader. Transducers with high directivity help to suppress ambient echoes, but they require precise alignment and limit the read area. The signals of the decaying oscillation of the resonator can be separated in the reader if the quality factor of the resonator is significantly higher than the equivalent quality factor of the power delay profile of the transmission channel.

Figure 4 shows schematically an input signal in the reader after the end of the stimulation signal as a function of time. The left diagram shows the real part of the sampled values and the diagram to the right shows both the real and imaginary part. For signal processing, only sampled values according to the sampling theorem are relevant, i.e., two samples per oscillation. For a better overview, 10 samples per oscillation are shown here and they are also connected. The first samples are dominated by stochastic ambient echoes, which, however, soon fade away. As the process progresses, the signals from the decaying resonator stand out from the noise. After about 10 oscillations, these usable signals disappear into the noise. The quality factor of the oscillator in this schematic drawing is 10 and the signal-to-noise ratio at t = 0 is 20 dB.

Ambient echoes would cause a systematic error in the evaluation of the resonance frequency, which cannot be eliminated later. Also, averaging would not lower this systematic error. Therefore, the ambient echoes must be eliminated by a time gate $t_{gate}$ before the resonator signal is analyzed by the reader unit.

Choosing this time gate is relevant. If the time gate is set too short, systematic measurement errors result. If it is set too long, only an unnecessarily decayed signal is available for evaluation and the measurement accuracy suffers. In some applications, the power delay profile of the transmission channel can be measured before deploying the passive transponder. A method for adjusting the time window is suggested in Section 4.4.

Due to the exponentially decaying oscillation, the time gate results in a further attenuation of the initial power $P_{in}$ that can be analyzed in the reading device to $P_{gated}$ with(13)$$P_{gated}=\frac{1}{2}\cdot P_{t}\cdot {g_{t}}^{2}\cdot {g_{r}}^{2}\cdot e^{-2\alpha t_{gate}}\cdot e^{-2\alpha_{m}R}\cdot {\left(\frac{\lambda }{4\pi R}\right)}^{4}\cdot {H_{R}}^{2}\left(\omega \right)\cdot {\left(1-e^{\left(j\omega_{d}-j\omega -\alpha \right)t_{load}}\right)}^{2}.$$

It describes the link budget when reading a cooperative resonant passive target by a radar system. The term ${\left|1-e^{\left(j\omega_{d}-j\omega -\alpha \right)t_{load}}\right|}^{2}$ describes the state of the loading of the resonator after the loading time $t_{load}$; ${H_{R}}^{2}$ represents the results of both the matching of the readout frequency ω to the natural frequency $\omega_{d}$ and the electrical matching of the impedance of the transducer to the resonator. ${\left(\frac{\lambda }{4\pi R}\right)}^{4}$ gives the geometrical spread of the readout signal and the response signal, $e^{-2\alpha_{m}R}$ a possible viscose damping of the signal along the distance *R* in the transmission medium, $e^{-2\alpha t_{gate}}$ the decaying of the resonator signal due to the waiting time $t_{gate}$ until all strong environmental echoes have died out, and $g_{t}$ and $g_{r}$ the gain of the transducers to couple into the transmission medium. If a bi-static setup is applied, ${g_{t}}^{2}$ or ${g_{r}}^{2}$ has to be replaced by $g_{t1}\cdot g_{t2}$ or $g_{r1}\cdot g_{r2}$, respectively.

The decay of the ambient echoes usually can be characterized by a quality factor, $Q_{tc}$. The above equation can be rewritten with the help of $Q_{tc}$ as(14)$$P_{gated}=e^{\left(-2\pi \frac{Q_{tc}}{Q}\right)}\cdot P_{in}.$$

Figure 5 shows the start of the received power $P_{gated}$, which is still present in the reader after the time gate in order to eliminate ambient echoes before the start of sampling. This power level then decreases according to the quality factor of the resonator. The excitation setup and the colors for the frequency bands were selected as in Figure 3. The remaining power for resonators with a quality factor of 10 is shown in dotted lines, for resonators with a quality factor of 100 in dashed lines, and for resonators with a quality factor of 1000 in solid lines. The quality factor of the transmission channel $Q_{tc}$ was set to 15. The signals from the resonators with a quality factor of 10 are strongly attenuated by the time gate, while the signals from the resonators with a quality of 100 or 1000 were hardly reduced.

The signal with power $P_{gated}$ in Equation (Equation 13) usually is boosted with a low-noise amplifier and then mixed down to baseband with the actual local oscillator signal and sampled. Depending on the distance between the stimulating frequency ω and the natural frequency of the resonator $\omega_{d}$, the down mixed signal only shows a few oscillations before it disappears into noise.

### 2.3. Noise Power

The thermal environment of the resonator has the temperature ϑ and therefore emits thermal noise. The noise power $N_{p}$ within the receiver bandwidth *B*, which is received in addition to the resonator signal, is given by $k_{B}\vartheta B$, where $k_{B}$ is the Boltzmann constant. The low-noise amplifier increases this noise power by the factor $F_{LNA}$. This results in a total noise power $N_{p}$ of(15)$$N_{p}=F_{LNA}k_{B}\vartheta B.$$

The carrier frequency of the Lorentz curve can only be determined correctly if the entire spectrum of the Lorentz curve is contained in the analyzed frequency range. Each windowing of the Lorentz curve changes the decay constant and also shifts the apparent carrier frequency. For example, if the carrier frequency is outside the resolved frequency band, the signal processing cannot determine the correct frequency. The receiver bandwidth *B*, thus, must be larger than the resonance bandwidth of the resonator to resolve the decaying resonance(16)$$B>\frac{f_{0}}{Q}.$$

A measurement bandwidth of approximately three to four times this minimum bandwidth would be optimal for a maximum signal-to-noise ratio. However, in order to reliably detect the shift in the center frequency, a larger bandwidth than this minimum usually must be sampled, for example $0.5\%\;\mathrm{to}\;1\%\;f_{0}$. In the ISM and SRD frequency bands, the bandwidth is restricted by legal requirements.

The receiver bandwidth relevant for the noise power results from the sampling frequency $f_{s}$ or the sampling interval *T* of the receiver(17)$$B=\frac{1}{2}f_{s}=\frac{1}{2T}.$$

In order to minimize the noise power in the sampled signal, the received signal is limited to the minimum bandwidth of $0.5f_{s}$ before sampling. If the bandwidth of the signal is lowered below the value given in Equation (Equation 17), we call the signal over-sampled. The over-sampled content of the signal contains no additional information, and we can reduce in the following analysis the sampling frequency to half of the bandwidth.

After data collection is complete, usually a new measurement can be started. If neither the measured value, i.e., the desired resonance frequency, nor the transmission channel changes, either the extracted resonance frequencies can be averaged or, in some cases, the individual complex sample values also can be averaged coherently to lower the equivalent noise bandwidth *B*.

If the noise can be described by a mean-free Gaussian process, it is completely characterized by the variance ${\sigma_{n}}^{2}$, where(18)$${\sigma_{n}}^{2}=N_{p}.$$

Since the amplitude of the exponentially decaying resonator signal decreases with time, the signal-to-noise ratio becomes time-dependent. However, we can define the initial signal-to-noise ratio η as the ratio of the initial signal power $P_{gated}$ to the noise floor $N_{p}$, which is given by(19)$$\eta =\frac{P_{gated}}{N_{p}}=\frac{P_{gated}}{{\sigma_{n}}^{2}}.$$

Figure 6 shows the initial signal-to-noise ratio η in the reader after the time gate for the same frequency ranges and resonator qualities as in Figure 5. The noise bandwidths used in the left graph were 1.75 MHz for the 433 MHz band, 2 MHz for the 868 MHz band, 26 MHz for the 915 MHz band, 100 MHz for the 2.45 GHz band and 150 MHz for the 5.8 GHz band. However, sampling the same relative bandwidths $B=0.5\%f_{0}$ results in the graph to the right with significant improvements in the signal-to-noise ratio at the wide bands 915 MHz, 2450 MHz and 5800 MHz.

### 2.4. Sampling

The results of the Crámer–Rao lower bound calculations depend on an agreement on the starting time of the samples, as this changes the phases. There are two special time frames: either the samples are taken symmetrical to t=0, or sampling starts at t=0. The second choice seems to best fit for an exponentially decaying signal. Therefore, we shift the time frame to start sampling at t=0. The analogue mixed down signal $s_{md}\left(t\right)$ is then given by(20)$$s_{md}\left(t\right)=\sqrt{P_{gated}}\cdot e^{\left(-\alpha +j\omega_{d}-j\omega \right)t+j\varphi_{3}},$$
with $P_{gated}$ given in Equation (Equation 13). Downconversion reduces the number of oscillations in the signal, but α is preserved.

If we neglect the phase shift due to the interference between the forced oscillation and the natural oscillation in the resonator, the phase shift $\varphi_{3}$ of the mixed down analog signal compared to the local oscillator of the reader is given by(21)$$\varphi_{3}=\left(\omega +\omega_{d}\right)\frac{R}{c}+\left(\omega -\omega_{d}\right)t_{gate}.$$

Since *R* and $\omega_{d}$ are unknown, $\varphi_{3}$ is also unknown. In order to simplify the following formulas a bit, we introduce the following abbreviations(22)$$\tilde{b}=\sqrt{P_{gated}},\;\;\tilde{\omega }=\left(\omega_{d}-\omega \right),\;\;\tilde{\varphi }=\varphi_{3}.$$

The exponentially decaying signal then can be written as(23)$$s_{md}\left(t\right)=\tilde{b}e^{\left(-\alpha +j\tilde{\omega }\right)t+j\tilde{\varphi }}.$$

This signal is completely described by four variables that can be combined into a state vector χ, which is given by(24)$$\chi =\left(\begin{array}{c} \tilde{\omega }\\ \tilde{\varphi }\\ \alpha \\ \tilde{b} \end{array}\right).$$

Two variables, $\tilde{\omega }$ and $\tilde{\varphi }$, describe the phase and two, α and $\tilde{b}$, the amplitude of the decaying phasor, with one variable each characterizing the change over time ($\tilde{\omega }$ and α) and one variable ($\tilde{\varphi }$ and $\tilde{b}$) the initial state. The sampling delivers the discrete-time signal $s_{n}$, where *n* is the index of an element.(25)$$s_{n}=s_{md}\left(t_{n}\right)=\tilde{b}e^{\left(-\alpha +j\tilde{\omega }\right)t_{n}+j\tilde{\varphi }}=\tilde{b}e^{\left(-\alpha +j\tilde{\omega }\right)n\cdot T+j\tilde{\varphi }},\;\;\mathrm{for}\;\;n=0,1\cdots N-1.$$

In addition to $s_{n}$, the sampled complex signal $z_{n}$ contains noise $n_{n}$. The average noise power is given by Equation (Equation 15). The sampled common signal $z_{n}$ is given by(26)$$z_{n}=s_{n}+n_{n}.$$

## 3. Crámer–Rao Lower Bounds for Exponentially Decaying Phasors

Due to the noise, only an estimate for χ can be determined. For any given estimate of the state vector χ, we can calculate the resulting estimates $\mu_{n}$ for the sampled signals $z_{n}$ at times n·T:(27)$$\mu_{n}=\tilde{b}e^{n\cdot \left(-\alpha +j\tilde{\omega }\right)T+j\tilde{\varphi }}.$$

### 3.1. Maximum Likelihood Estimation of χ and $\sigma_{n}$

If the noise is Gaussian distributed, with mean $\mu_{n}$ and variance ${\sigma_{n}}^{2}$, the probability density function $f\left(z_{n};\chi \right)$ of a single sample point can be modeled as(28)$$f\left(z_{n};\chi \right)=\frac{1}{2\pi {\sigma_{n}}^{2}}e^{-\frac{1}{2{\sigma_{n}}^{2}}{\left|z_{n}-\mu_{n}\right|}^{2}}.$$

$f\left(z_{n};\chi \right)$ gives the probability of measuring $z_{n}$ if $\mu_{n}$ was the true value. Obviously, $f\left(z_{n};\chi \right)$ is maximal when $\mu_{n}=z_{n}$. Now, we adapt our analytical signal, which is described by the four parameters in the state vector χ, to the measurement points in such a way that the deviation of the measurement points from the analytic signal corresponds to the narrowest possible Gaussian distribution. This is accomplished by multiplying all probability density functions of all linearly independent samples in the measurement vector, yielding the joint probability density function f(z;χ), also called the likelihood function, and finding the global maximum of this function,(29)$$f\left(z;\chi \right)=\prod_{n=0}^{N-1}f\left(z_{n};\chi \right)={\left(\frac{1}{2\pi {\sigma_{n}}^{2}}\right)}^{N}e^{-\frac{1}{2{\sigma_{n}}^{2}}\sum_{n=0}^{N-1}{\left|z_{n}-\mu_{n}\right|}^{2}}.$$

The assumption that the noise is linearly independent for each sample and for the real and imaginary components applies when the sampling frequency corresponds to the Nyquist frequency and when interference from spurious reflections and crosstalk between I and Q channels are negligible.

The maximum stays the same if we analyze the logarithm of Equation (Equation 29), since the logarithm is a monotonic function. This results in the so-called log-likelihood function(30)$$\log\left(f\left(z;\chi \right)\right)=-N\log\left(2\pi {\sigma_{n}}^{2}\right)-\frac{1}{2{\sigma_{n}}^{2}}\sum_{n=0}^{N-1}{\left(z_{n}-\mu_{n}\right)}^{*}\left(z_{n}-\mu_{n}\right).$$

The best estimate of the variables in χ maximizes the probability that our measurements will turn out as we obtain them. In this case, they maximize f(z;χ) in Equation (Equation 29) or (Equation 30) for a given measurement $z_{n}$.

Of the five variables in the likelihood function, $\tilde{b}$, α, $\tilde{\omega }$, $\tilde{\varphi }$ and $\sigma_{n}$, three ($\tilde{b}$, $\tilde{\varphi }$ and $\sigma_{n}$) can be calculated analytically from Equation (Equation 30) once α and $\tilde{\omega }$ are determined (see Appendix B).

For $\tilde{\varphi }$ we get (see Section B.1, Equation (A43)):(31)$$\tilde{\varphi }=\arg\sum_{n=0}^{N-1}z_{n}e^{-\alpha nT}e^{-j\tilde{\omega }nT}$$

After multiplication of the measurement points by the phase factor $e^{-j\tilde{\omega }nT}$, which removes the carrier frequency, all measuring points are distributed in a Gaussian shape around a certain phase angle. The best estimate for the initial phase $\tilde{\varphi }$ is then the phase angle of the weighted sum of all measurement points without carrier frequency multiplied by the associated amplitude of the decaying resonator $e^{-\alpha nT}$. This reduces the impact of noise, especially in the low useful signal region.

For $\tilde{b}$, we obtain (see Section B.2, Equation (A44)):(32)$$\tilde{b}=\frac{1-e^{-2\alpha T}}{1-e^{-2N\alpha T}}\cdot \left|\sum_{n=0}^{N-1}z_{n}e^{-\alpha nT}e^{-j\tilde{\omega }nT}\right|$$

We remove the carrier frequency from the measurement points, multiply it by the associated amplitudes of the decaying resonator signal, analogous to a matched filter, and sum up all values. We divide the result by the analytically calculated sum of the sampled powers of a decaying resonator of the same quality factor that begins with amplitude 1.

Using the estimates for χ, we can calculate an estimate of the standard deviation of the Gaussian noise using the mean squared error between the measured signal and the estimates (see Section B.3, Equation (A47)).(33)$${\sigma_{n}}^{2}=\frac{1}{2N}\sum_{n=0}^{N-1}{\left|z_{n}-\mu_{n}\right|}^{2}$$

However, to solve Equations (Equation 31), (Equation 32) and (Equation 33), the variables α and $\tilde{\omega }$ must be known. If we substitute Equation (Equation 31) and (Equation 32) into Equation (Equation 30), we obtain two remaining optimization equations with respect to $\tilde{\omega }$ and α (see Equation (A45))(34)$$\frac{\partial }{\partial \tilde{\omega },\alpha }\left[\left|\sum_{n=0}^{N-1}z_{n}e^{-\alpha nT}e^{-j\tilde{\omega }nT}\right|\right]=0$$

The above equation is unsuitable for analytical or numerical optimization. A well-suited approximation method for solving is discussed in Section 4.

### 3.2. Crámer–Rao Lower Bounds

Due to the finite sample with N noisy values, the parameters in χ can only be determined with a finite precision, the so-called asymptotic Crámer–Rao lower bounds. This case is analogous to the uncertainty in a mean value, which is calculated from a finite number of noisy constant measurement values.

The Crámer–Rao lower bounds are calculated using the Fisher information matrix *J*. In the case where we have additive white Gaussian noise, unbiased complex data and unknown real parameters in χ, the Fisher information matrix can be calculated using [50] (p. 49).(35)$$J_{ij}\left(\chi \right)=\frac{2}{N_{p}}R\left\{\sum_{n=0}^{N-1}\left[\frac{\partial {\mu_{n}}^{*}}{\partial \chi_{i}}\frac{\partial \mu_{n}}{\partial \chi_{j}}\right]\right\},$$
with $R\left\{x\right\}$ being the real part of *x*. Appendix A shows the calculation of the Fisher information matrix for the parameters in χ as well as its inverse matrix. The asymptotic Crámer–Rao lower bounds for the unknown parameters in χ are obtained from the main diagonal of $J^{-1}\left(\chi \right)$. We obtain (see Equations (A37) and (A38)).(36)$$\begin{array}{ccc} {\sigma_{\tilde{\omega }}}^{2} & = & {\sigma_{\alpha }}^{2}\\ & \geq & \frac{\left(1-e^{-2\alpha NT}\right){\left(1-e^{-2\alpha T}\right)}^{3}}{{2T}^{2}\eta \left[-N^{2}e^{-2\alpha NT}{\left(1-e^{-2\alpha T}\right)}^{2}+e^{-2\alpha T}{\left(1-e^{-2\alpha NT}\right)}^{2}\right]} \end{array}$$(37)$$\begin{array}{ccc} \frac{{\sigma_{\tilde{b}}}^{2}}{{\tilde{b}}^{2}} & = & {\sigma_{\tilde{\varphi }}}^{2}\\ & \geq & \frac{{1-e}^{-2\alpha T}}{2\eta }\left[1+\frac{-e^{-\alpha \left(4N+2\right)T}+\left(2N-1\right)e^{-\alpha \left(2N+4\right)T}+\;e^{-\alpha \left(2N+2\right)T}+e^{-4\alpha T}}{-N^{2}e^{-2\alpha NT}{\left(1-e^{-2\alpha T}\right)}^{2}+e^{-2\alpha T}{\left(1-e^{-2\alpha NT}\right)}^{2}}\right]. \end{array}$$

Since ω is known exactly, the variance ${\sigma_{\tilde{\omega }}}^{2}$ is identical to the variance ${\sigma_{\omega_{d}}}^{2}$. The variances ${\sigma_{\omega_{d}}}^{2}$ and ${\sigma_{\alpha }}^{2}$ can be rewritten as(38)$${\sigma_{\omega_{d}}}^{2}={\sigma_{\alpha }}^{2}\geq \frac{{\left(1-e^{-2\alpha T}\right)}^{3}}{{2T}^{2}e^{-2\alpha T}\eta }\cdot r\left(\alpha NT\right),$$
where $r\left(\alpha NT\right)$ is the residual of the above approximation with(39)$$r\left(\alpha NT\right)=\frac{e^{-2\alpha T}\cdot \left(1-e^{-2\alpha NT}\right)}{\left[-N^{2}e^{-2\alpha NT}{\left(1-e^{-2\alpha T}\right)}^{2}+e^{-2\alpha T}{\left(1-e^{-2\alpha NT}\right)}^{2}\right]}.$$

### 3.3. Limit Value for the Frequency Resolution $\sigma_{\tilde{\omega }}$

Sampled signals with a finite spectrum that satisfy the Nyquist criterion are completely determined, and the parameters of the signal, such as the carrier frequency, can therefore be estimated without errors. If a noise component is added to the signal, the signal can no longer be separated from the noise even if the Nyquist criterion is met and the carrier frequency can only be estimated with a variance that depends on the signal-to-noise ratio. If the noise is Gaussian distributed and bias-free, the minimum variance approaches zero as the number of sample points approaches infinity. In this case, the signal can be completely separated from the noise component. However, with sampled, exponentially decaying phasors, the measurement accuracy approaches a finite limit value even with an infinite number of sampling points, since the useful signals disappear into the noise after a few time constants of the decay.(40)$$\begin{array}{ccc} \lim_{N\to \infty }r\left(\alpha NT\right) & = & \frac{e^{-2\alpha T}}{e^{-2\alpha T}}=1 \end{array}$$(41)$$\begin{array}{ccc} \lim_{N\to \infty }{\sigma_{\omega_{d}}}^{2}=\lim_{N\to \infty }{\sigma_{\alpha }}^{2} & \geq & \frac{{\left(1-e^{-2\alpha T}\right)}^{3}}{{2T}^{2}e^{-2\alpha T}\eta } \end{array}$$(42)$$\begin{array}{ccc} \lim_{N\to \infty }\frac{{\sigma_{\tilde{b}}}^{2}}{{\tilde{b}}^{2}}=\lim_{N\to \infty }{\sigma_{\tilde{\varphi }}}^{2} & \geq & \frac{1-e^{-2\alpha T}}{2\eta }\left[1+\frac{e^{-4\alpha T}}{e^{-2\alpha T}}\right]=\frac{1-e^{-4\alpha T}}{2\eta } \end{array}$$

If the measurement cycles of excitation of the resonator, scanning and evaluation of the decaying resonator signal are repeated several times, a measurement accuracy below any specified lower limit can still be achieved by averaging the results.

The residual $r\left(\alpha NT\right)$ very quickly approaches the limit value 1, as Figure 7 shows. As a result, the variance $\sigma_{\omega_{d}}^{2}$ reaches its final value when the received signal is sampled along *Q* oscillations. In this case, αNT=π and r(π)≈1.

Figure 7 shows that the limit value specified in Equation (Equation 40) is nearly reached when αNT=π, i.e., after sampling of Q oscillations. $\sigma_{\omega_{d}}^{2}$ increases significantly, or the resolution decreases significantly, when fewer than Q oscillations are sampled. However, a longer sampling does not increase the resolution but rather costs time that could possibly be used for another measurement cycle and subsequent averaging. In addition, it increases the noise energy in the signal, while the energy of the useful signal remains almost constant. There are two ways to solve this problem: either the signal is truncated after the time $0.85\ldots 1.0\cdot Q/f_{d}$ or the signal is weighted with truncation as a special case of weighting.

An analysis in the Appendix C shows that an excitation and sampling time of $0.85\ldots 0.9\cdot Q/f_{d}$ with a subsequent restart of the measurement and an averaging of the measurement results promises the lowest variance $\sigma_{\omega_{d}}^{2}$. The literature gives also $0.9\cdot Q/f_{d}$ as the optimal length between improving resolution and reducing the impact of noise for a single signal to be analyzed [39]. However, the minimum toward longer stimulation and sampling times is very flat; the reduction in variance for a sampling time of $0.85\cdot Q/f_{d}$ is less than 1% compared to stimulation and sampling of $1.0\cdot Q/f_{d}$. After complete loading and sampling, the oscillation can reduce the ripple in the spectrum that would be caused by a hard on/off cycle.

### 3.4. Relative Frequency Accuracy by a Wireless Readout of a High-Q Resonator

If we substitute the signal-to-noise ratio in Equations (Equation 19) and (Equation 13) into Equation (Equation 36) or (Equation 38), we obtain the Crámer–Rao lower bound for the natural frequency.(43)$$\begin{array}{ccc} {\sigma_{\omega_{d}}}^{2}={\sigma_{\alpha }}^{2} & \geq & \frac{{\left(1-e^{-2\alpha T}\right)}^{3}}{{2T}^{2}e^{-2\alpha T}}\cdot \frac{1}{\eta }\cdot r\left(\alpha NT\right),\;\mathrm{with}\; \end{array}$$(44)$$\begin{array}{cccc} \; & \frac{1}{\eta } & = & \frac{F_{LNA}k_{B}\vartheta f_{s}}{P_{t}\cdot {g_{t}}^{2}\cdot {g_{r}}^{2}\cdot e^{-2\alpha t_{gate}}\cdot e^{-2\alpha_{m}R}{H_{R}}^{2}\left(\omega \right)\cdot {\left(1-e^{\left(j\omega_{d}-j\omega -\alpha \right)t_{load}}\right)}^{2}}\cdot {\left(\frac{4\pi R}{\lambda }\right)}^{4} \end{array}$$
and $r\left(\alpha NT\right)$ given in Equation (Equation 39). The relative frequency error $\sigma_{f_{0}}/f_{0}$ results from the above equations to(45)$$\frac{\sigma_{f_{0}}}{f_{0}}=\frac{1}{4\pi^{2}f_{0}}\cdot \sigma_{\omega_{d}}$$

Figure 8 shows the Crámer–Rao lower bound for the relative frequency error $\sigma_{f_{0}}/f_{0}$ that can be achieved in a single measurement for some selected ISM and SRD bands. For the left diagram, the maximum bandwidth in the frequency bands was used; in the right picture, a measurement bandwidth of $0.5\%f_{0}$ was used. Resonators with quality factors of 10, 100 and 1000 were used in each frequency range.

The bandwidths of the resonators with a quality factor of 10 were wider than the permitted frequency bands and, therefore, these resonators were not taken into account. The frequency bands 433 MHz and 868 MHz were also too narrow for the resonators with a quality factor of 100. With a measurement bandwidth of $0.5\%f_{0}$, only the resonators with quality factor of 1000 could be read.

To analyze the influence of the quality factor on the lower Crámer–Rao bound for the relative frequency error $\sigma_{f_{0}}/f_{0}$, a sweep on *Q* in Equation (Equation 45) was performed. Figure 9 shows the reduction in relative frequency error for one single readout cycle with increasing resonator quality factor. A readout distance R=1λ was assumed for the illustration. The displayed $\sigma_{f_{0}}/f_{0}$ must be multiplied by ${\left(R/\lambda \right)}^{2}$ in order to scale to a different read distance *R*. The diagrams start with the quality factor at which the resonance bandwidth fits into the ISM or SRD band. For a quality factor where the resonator bandwidth became smaller than one fifth of the assigned bandwidth, the sampled bandwidth was fixed at five times the resonator bandwidth.

Equations (Equation 38) and (Equation 39) can be further simplified if *Q* oscillations of the RF signal or $Q\cdot B/f_{0}$ oscillations in the base band are sampled. In this case, we obtain αNT=π and r(αNT)≈1:(46)$${\sigma_{\omega_{d}}}^{2}\geq \frac{{\left(1-e^{-2\alpha T}\right)}^{3}}{{2T}^{2}e^{-2\alpha T}}\cdot \frac{1}{\eta }$$

Above approximation is shown by dotted lines in Figure 9.

## 4. Calculating the Natural Frequency of a Decaying Oscillator

Equations (Equation 29) or (Equation 30) contain five unknowns, the four parameters of χ and ω. Three of them, $\tilde{\varphi }$, $\tilde{b}$ and ω, can be determined from Equation (Equation 30) using the maximum likelihood method once $\tilde{\omega }$ and α are known. However, $\tilde{\omega }$ and α must be determined numerically. Equations (Equation 29), (Equation 30), or (Equation 34) are completely unsuitable for numerical optimization.

The fastest and computationally most effective methods for accurately estimating the frequency of decaying exponentials in noise operate in the frequency domain and use the fast Fourier transformation (FFT). Rife and Boorstyn showed that the maximum likelihood frequency estimator for the decaying resonator signal in noise is given, just as in the case of an undamped sine wave, using the argument of the periodogram maximizer, i.e., looking for the parameter λ which provides the maximum in the discrete Fourier transform [50,51]:(47)$${\hat{f}}_{ML}=\underset{\lambda }{arg max}\left\{Y\left(\lambda \right)\right\},\;\mathrm{where}\;\;Y\left(\lambda \right)=\left|\sum_{n=0}^{N-1}w_{n}e^{-j2\pi n\lambda }\right|$$

To determine the carrier frequency and damping constant of a decaying oscillator, a two-stage procedure is proposed in the literature, which is analogous to the case of a constant oscillator oscillation. In a first search, the rough frequency position is narrowed down and, based on this, the exact frequency is determined in the second step.

### 4.1. Coarse Search for the Carrier Frequency

The coarse position of the carrier frequency is usually determined using the Fourier transformation. In Appendix C, it was found that a sampling over approximately Qf oscillations with a subsequent new start of the measurement and subsequent averaging of the complex samples in the time domain promises optimal resolution for a given measurement time. In this case, without zero padding, there is a step size of fQ in the frequency domain, i.e., a fairly coarse spectral resolution.

Since always Qf oscillations of a decaying exponential function with noise are sampled, the waveforms are similar in the time domain. Therefore, the main lobes are also similar in the frequency domain. Figure 10 shows on the left the distribution of the sampling points around the maximum as small blue circles when calculating the Fourier transform without zero padding. The red line was calculated with a zero padding of 10.

If the measurement signal is demodulated with the frequency of the maximum of the sampling points, which were calculated without zero padding via the Fourier transformation, a residual oscillation of up to ±12 period remains. Figure 10 shows on the right the measured values in magnitude and phase that were demodulated with this coarse value for the carrier frequency. Since the actual frequency is about a quarter step size away from the maximum, a phase error of approximately 2π4=90∘ remains. The noise increases significantly when the signal from the decaying resonator slowly disappears into the noise.

### 4.2. Fine Search for the Carrier Frequency

Without zero padding, the width of the main lobe in the frequency domain is about two step sizes, and the points to the left and right of the maximum are approximately at the height of the 3 dB points. As a result, the Lorentz curve around the resonance frequency is quite poorly resolved for an exact determination of the maximum. In addition, the Lorentz curve can not be approximated with a parabolic approximation using the three points around the maximum, which is to be expected from Figure 10 on the left, but it can be seen clearly in Figure 11.

To achieve a resolution close to Crámer–Rao’s lower bound, we need to fit the sample values in the frequency domain around the maximum with a Lorentz curve. In [39,40], Elias Aboutanios presents an iterative algorithm for the extremely efficient and highly accurate calculation of the maximum of an approximated Lorentz curve. The algorithm uses two DFT values that must have a mutual frequency separation of 1 FFT step size and that must be above and below the maximum. In Appendix E, we deduce the algorithm.

The discrete Fourier coefficients $S_{k}$ in the frequency domain are calculated from the sample values $s_{n}$ with(48)$$S_{k}=\sum_{n=0}^{N-1}s_{n}e^{-j2\pi \frac{nk}{N}}$$

Let *m* be the index of the maximum of the Fourier coefficients, corresponding to a frequency $f_{old}$, with(49)$$f_{old}=\frac{m}{NT}.$$

Then, we need the DTF components for $k=m\pm \frac{1}{2}$. Half-step sizes in the frequency domain can be achieved either by zero padding N points before calculating a FFT or by evaluating the DFT, Equation (Equation 48), for $k=m\pm \frac{1}{2}$. We obtain $S_{m-\frac{1}{2}}$ and $S_{m+\frac{1}{2}}$, and we define(50)$$h=\frac{1}{2}\frac{S_{m+\frac{1}{2}}+S_{m-\frac{1}{2}}}{S_{m+\frac{1}{2}}-S_{m-\frac{1}{2}}}$$

An improved estimate of the resonance frequency $f_{updated}$, as well as an estimate of the damping constant α, is then given by(51)$$\begin{array}{ccc} f_{updated} & = & f_{old}+\frac{R\left\{h\right\}}{NT} \end{array}$$(52)$$\begin{array}{ccc} \alpha & = & \frac{2\pi }{NT}I\left\{h\right\}, \end{array}$$
with $R\left\{h\right\}$ and $I\left\{h\right\}$ being the real and imaginary part of *h*.

Based on this improved estimate for the resonance frequency, we can now use the discrete Fourier transformation (DTF) in Equation (Equation 48) to calculate the two Fourier coefficients half a step size from the improved estimate to the left and right(53)$$S_{m+R\left\{h\right\}\pm \frac{1}{2}}=\sum_{n=0}^{N-1}s_{n}e^{-j2\pi \frac{n\left(m+R\left\{h\right\}\mp \frac{1}{2}\right)}{N}}$$
and thus run the optimization again.

Figure 12 shows three examples of the convergence of Aboutanio’s algorithm (Equations (Equation 50) and (Equation 51)) for three different signal-to-noise ratios. For the simulation, we calculated the complex values in Matlab [43] for a phasor with a center frequency of 1, a quality factor of 100.23 to avoid a period in the sampled signal, and a given starting phase. The sampling times were chosen to $t_{i}=i\cdot 0.35,i=1\ldots \;N$ until $t=1.0633\cdot Q/f_{0}$ is reached, which defines *N*. We then added complex noise according to the signal-to-noise level under study. A new noise signal was generated with each run. The following fast Fourier transform was performed with a zero padding of N points. The frequency of the maximum of the amplitude was taken as the first estimate of the natural frequency, and the values to the left and right of it were used for the improvement algorithm in Equations (Equation 50) and (Equation 51). The resulting improved natural frequency was taken as the new center, and the values of the left and right frequency points half a FFT step size apart were calculated using the DFT in Equation (Equation 48). This was repeated five times, and the updated natural frequencies were plotted along with ±1σ of the Crámer–Rao lower bound, which is shown in dashed lines. The estimated center frequencies start on the left with the frequency of the sampling point of the maximum of the spectrum calculated with a zero padding of N points and then converge very quickly to the best estimate for that dataset. According to the definition shown in Equation (Equation 19), the signal-to-noise ratio is 20 dB for the left graph, 10 dB for the middle one and 0 dB for the right one. Ten examples are shown for each signal-to-noise value. Note the different scaling of the ordinates of the graphs.

Figure 13 shows the histograms for the simulations shown in Figure 12 for 2000 runs each. The signal-to-noise ratios again are 20 dB (left), 10 dB (middle) and 0 dB (right). In addition, the normalized Gaussian of the corresponding Crámer–Rao lower bound is shown. Depicted is an area of ±6σ of the Crámer–Rao lower bound, which leads to a different abscissa scaling.

The histograms show a somewhat broader distribution function than the analytical Gaussian distributions; in particular, there are missing entries near the mean value, but there is a clear excess of entries in the outer ranges. This behavior can be seen very clearly in the comparison of the cumulative density functions in Figure 14. The signal-to-noise ratios again are 20 dB (left), 10 dB (middle) and 0 dB (right).

### 4.3. Weighting for Improving the Signal-to-Noise Ratio

The algorithm works correctly even with low-quality factors, as long as the maximum in the frequency domain is dominated by the signal of the decaying resonator. If noise, interference, and aliasing signals become too strong, a noise peak might overpower the resonant signal and, therefore, lead to incorrect results. One way to maximize the signal to noise energy is to cut off the sampled signal at $Q/f_{d}$ or to stop sampling there.

The amplitude of the spectrum exactly at resonance frequency is (see Equation (A52))(54)$$S\left(f_{0}\right)=\sum_{n=0}^{N-1}s_{n}=\tilde{b}e^{j\tilde{\varphi }}\frac{1-e^{-\alpha NT}}{1-e^{-\alpha T}}.$$

The exponential function in the numerator is very small and so is the exponent in the denominator. The amplitude at resonance frequency can therefore be approximated as(55)$$|S\left(f_{0}\right)|\approx \frac{\tilde{b}}{\alpha T}=\frac{\tilde{b}Q}{\pi f_{0}T}=\frac{\tilde{b}Q}{2\pi }.$$

Figure 15 shows the detection probability of the algorithm as a function of the signal-to-noise ratio with 5000 simulations each. A correct detection was assumed if the calculated natural frequency was within an interval of ±6σ around the actual natural frequency. The left chart is calculated for a resonator quality factor of 10, the middle for 100 and the right for 1000.

The curves do not scale, as suggested in Equation (Equation 55), with $Q^{2}\cdot \eta$, i.e., the same detection probabilities arise for the same $Q^{2}\cdot \eta$ product but with $Q^{0.75}\cdot \eta$. This effect is caused by the usage of different sample lengths chosen to $Q/f_{d}$. Thus, the noise energy scales from one graph to another by a factor of 10. If we take this into account, the curves scale with $\;Q^{1.75}\cdot \eta$, which is quite similar to Equation (Equation 55).

The curves do not go to zero even at a very low signal-to-noise ratio, especially on the graph with a quality factor of 10, as there remains a finite probability that a maximum noise peak occurs within ±6σ around the actual resonance frequency. For $Q^{0.75}\cdot \eta \approx 6$, the detection probability reaches 50%, and for $Q^{0.75}\cdot \eta \geq 50$, it approaches to 100%.

Figure 16 shows the convergence of 10 runs at signal-to-noise ratios where the detection probabilities are only 80%. In the left diagram, the signal-to-noise ratio is 3.9 dB for signals from resonators with a quality factor of 10, in the middle −4.2 dB for a resonator with a quality factor of 100 and on the right −12.85 dB for a resonator with a quality factor of 1000. Typically, eight trajectories can be seen. In dashed lines, ±1σ of the Crámer–Rao lower bound is shown.

The convergence of the algorithm decreases as the length of the sampling length increases above $Q/f_{d}$. Figure 17 shows on the left the convergence as a function of the sample length for every 5000 runs. The quality factor of the resonator is 100 and the initial signal-to-noise ratio is 0 dB. For this combination, the middle graph in Figure 15 shows a detection probability of 100% for a sampling length of $1\cdot Q/f_{d}$. The sample length here is increased from $1\cdot Q/f_{d}$ to $20\cdot Q/f_{d}$, which reduces the probability of success from 100% to 20%.

There are still some hits within $1\sigma_{f_{d}}$, but only 20%. Figure 17 shows on the right the histogram around the correct natural frequency for 5000 runs with a sample length of $20\cdot Q/f_{d}$. The 1σ probability distribution of the Crámer–Rao lower bound is also shown.

The impact of noise can be significantly reduced by weighting the signal in the time domain with an exponentially decaying window, which is similar to Equations (Equation 31) and (Equation 32) [52], see also Appendix D. As a result, the noise terms become correlated but remain mean-free.

For this purpose, in a first step, the noisy natural frequency $f_{d}$ and damping term α are determined from the unweighted time domain signal, and in a second analysis step, the measurement signal $z_{n}$ is weighted with the function $e^{-n\cdot \gamma \cdot \alpha T}$, resulting in $w_{n}$. Using this new time domain signal, the natural frequency and damping constant are determined again. Aboutanios gives for the weight factor γ an approximation [52](56)$$\begin{array}{ccc} w_{n} & = & z_{n}e^{-n\cdot \gamma \cdot \alpha T},\;\;\mathrm{with}\\ \gamma & = & \left\{\begin{array}{c} 0.02{\left(\alpha NT\right)}^{2}+0.02\left(\alpha NT\right)+0.39\;\;\mathrm{for}\;\;\;0<\left(\alpha NT\right)\leq 3\\ 1-1.08e^{-0.41\left(\alpha NT\right)}-0.07e^{-0.08\left(\alpha NT\right)}\;\;\;\mathrm{for}\;\;\;\left(\alpha NT\right)>3 \end{array}\right.. \end{array}$$

Figure 18 shows γ on the left as a function of αNT. When this weighting function is applied to the signal used in the center plot of Figure 17, i.e., on the decay signal of a resonator with a quality factor of 100 and an initial signal-to-noise ratio of 0 dB, the detection probability remains constant at 100% for all sample lengths from $1\cdot Q/f_{d}$ to $50\cdot Q/f_{d}$.

Even for the sample length of $1\cdot Q/f_{d}$, the weighting improves the detection probability at low signal-to-noise ratios, as can be seen in the right diagram of Figure 18. The red line is identical to the middle curve in Figure 15 and shows the detection probability of the algorithm for an unweighted signal as a function of the signal-to-noise ratio for a resonator signal with a quality factor of 100. The blue line uses the same signals but applies the weighting function of Equation (Equation 56). This shifts the detection probability by approximately 3 dB to lower signal-to-noise ratios.

If the measurement data are weighted according to the weight function in Equation (Equation 56), the resulting distribution of the simulation results corresponds to the Gaussian functions of the Crámer–Rao lower bound. Figure 19 shows the cumulative distribution functions along with the corresponding Gaussian functions for 5000 simulations of a resonator signal with a quality of 100 and signal-to-noise ratios of 20 dB (left), 10 dB (middle), and 0 dB (right). The curves are so close together that they can hardly be distinguished.

### 4.4. Detection of the Ambient Echoes of the Transmission Channel

After the natural angular frequency $\tilde{\omega }$ and the decay constant α are determined using Equations (51) and (52), the phase angle $\tilde{\varphi }$ and the amplitude $\tilde{b}$ at t=0 can be calculated by Equations (31) and (32). For this purpose, the natural frequency in the individual sampling points $z_{n}$ is eliminated by multiplying with $e^{-j\tilde{\omega }nT}$, and the influence of noise is reduced by weighting with $e^{-\alpha nT}$. All data are then added up. The phase of this sum gives the phase angle $\tilde{\varphi }$ at t=0. The amplitude $\tilde{b}$ at t=0 is obtained by normalizing the amplitude of this sum with the amplitude of a corresponding geometric series with a starting amplitude of 1. Figure 20 shows on the left the phase angles of the individual sampling points $z_{n}$ with the natural frequency eliminated and weighted with $e^{-\alpha nT}$. In addition, the phase angle $\tilde{\varphi }$ is displayed. The middle graph shows the associated amplitudes of the individual sample points $z_{n}$ on a logarithmic scale along with $\tilde{b}$.

After all four variables of the state vector χ have been determined, and the resulting estimates $\mu_{n}$ for the sampled signals $z_{n}$ at times n·T can be calculated according to Equation (Equation 27) and with these, using Equation (Equation 33), the noise components together with the echoes in the transmission channel. Figure 20 shows on the right in red the noisy signal of the decaying resonator with a signal-to-noise ratio of 20 dB together with the echoes in the transmission channel. In order to eliminate these echoes, the signal was only evaluated from t=0. For the curve shown in blue, the estimated values $\mu_{n}$ were subtracted from the measurement values $z_{n}$. The ambient echoes and noises remain.

Since the echoes in the transmission channel are systematic and do not change between individual readings, they add up coherently. If several measurement signals are summed up, in which the estimated values of the decaying resonator have been subtracted, the echoes of the transmission channel stand out clearly from the noise. Figure 21 shows the coherent sum of 50 signals with subtracted estimated values, on the left in linear scale and on the right in logarithmic scale. The systematic interference signals disappear into the noise at t=−15, and the signal evaluation could be started earlier by this interval, resulting in a higher resolution.

## 5. Summary and Outlook

The accuracy of a remote readout of a passive resonator depends sensitively on the initial signal-to-noise and interference ratios as well as the decay constant of the signal. This manuscript is limited to linear, time-invariant transmission channels without interference from other transmitters, linear, time-invariant transponders with constant resonance frequency, and Gaussian distributed, bias-free measurement uncertainties. For such a system, we showed the dependence of the initial signal-to-noise and interference ratio on the radar parameters’ transmission power, gain of the transducer to the transmission medium, reading distance and stimulation duration as well as on the passive transponder parameters’ resonance frequency, quality factor and impedance matching of the resonator to the transducer. The equations for the maximum likelihood estimation of the parameters of the exponentially decaying discrete-time signal and their Crámer–Rao lower bounds were presented. A stimulation and sampling time of approximately $Q/f_{d}$ for a single remote reading of the passive resonator promises the highest measurement resolution when subsequently averaged over multiple measurements. An iterative numerical algorithm enables an accuracy in determining the resonator parameters’ natural frequency and decay constant down to the Crámer–Rao limit. With these parameters, the systematic interference caused by echoes in the transmission medium could be determined.

The algorithm described in this manuscript allows to minimize the statistical errors in a wireless reading of a passive resonant transponder. Systematic errors, such as a pull in the resonance frequency due to external capacitive or inductive loads on the transducer, interference caused by stochastic jamming, shadowing or crosstalk in the reader, and time-varying transmission channels due to movements or rotations must be evaluated using more advanced algorithms [27,29,30].

The presented analysis allows a comparison between the theoretically possible measurement accuracy and an actual one of any experimental or commercial system based solely on the physical and electrical parameters of the measurement system. This comparison allows the qualitative and quantitative assessment of the performance of the system and shows any potential for optimization. It also allows the prediction of the measurement accuracy which can be expected at a given distance or what minimum resonator quality is required for a given measurement distance and a given measurement accuracy.

## Figures and Tables

**Figure 1 sensors-25-00747-f001:**
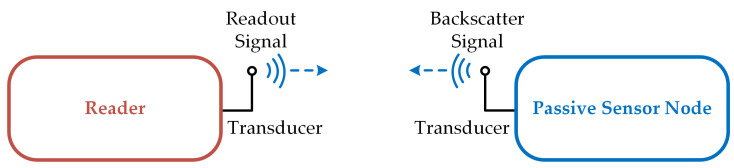
Schematic of the operation of a transponder system using passive resonant sensors: A reader sends a read signal (blue arrow to the right) to the sensor node over a wireless channel. This signal is received there and then stored in an oscillation in the resonator. When the read signal is switched off, the oscillation resonates with an exponentially decreasing amplitude at a frequency, which is modified by the quantity to be measured. A part of the stored energy is sent back to the reader as a backscatter signal (blue arrow to the left). There, it is received, sampled and evaluated.

**Figure 2 sensors-25-00747-f002:**
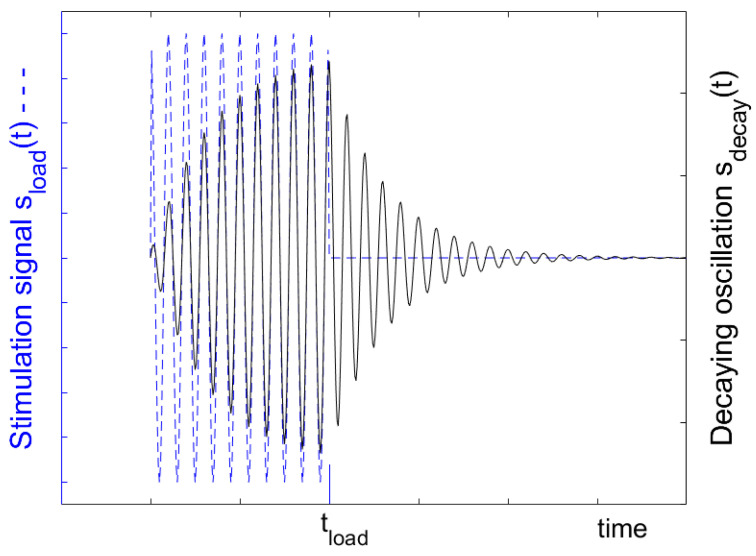
Schematic representation of the stimulation signal $s_{load}\left(t\right)$ delivered from the antenna to the resonator and the stored oscillation $s_{decay}$ in the resonator. The amplitude and frequency of the excitation signal were normalized to one, which is the time to $t\cdot f_{0}$. The resonator in this example has a quality factor of 10. Shown is the real part of $s_{load}\left(t\right)$ and $s_{decay}$. The stimulation signal stops at $t_{load}$, and the oscillation decays. Half of the signal from the decaying resonator is applied to the source impedance of the antenna, the other half to the internal resistance of the resonator. The figure is taken from [49].

**Figure 3 sensors-25-00747-f003:**
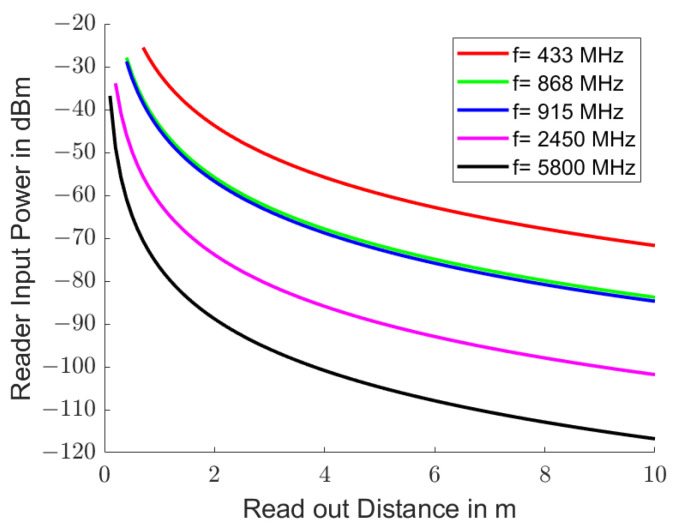
Typical reception power $P_{in}$ in the reader when reading passive resonant transponders. Electromagnetic propagation in free space was assumed ($\alpha_{m}=0$). The transmission power $P_{t}$ is 10 mW, the antennas radiate into the half-space ($g_{t}=g_{r}=2$) and the stimulation was carried out at resonance frequency ($H_{R}\left(\omega \right)=1$) over the duration of *Q* oscillations ($\alpha \cdot t_{load}=\pi$). The curves are drawn from top to bottom for the frequencies 433 MHz (red line), 868 MHz (green line), 915 MHz (blue line), 2.45 GHz (magenta line) and 5.8 GHz (black line) starting from a distance of 1 λ.

**Figure 4 sensors-25-00747-f004:**
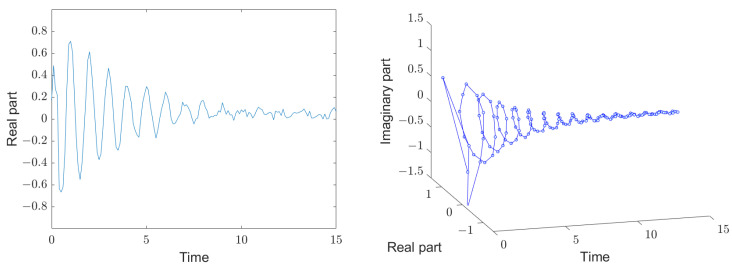
Schematic representation of the complex input signals of the reader, real part on the left and real and imaginary parts on the right, both as a function of time. Frequency and time are scaled as in Figure 2. Initially, the ambient echoes are larger than the resonator’s decaying signal, but they have a smaller quality factor and soon disappear. In the next phase, the resonator signals dominate, which later also disappear into the noise. The resonator in this schematic has a quality factor of 10.

**Figure 5 sensors-25-00747-f005:**
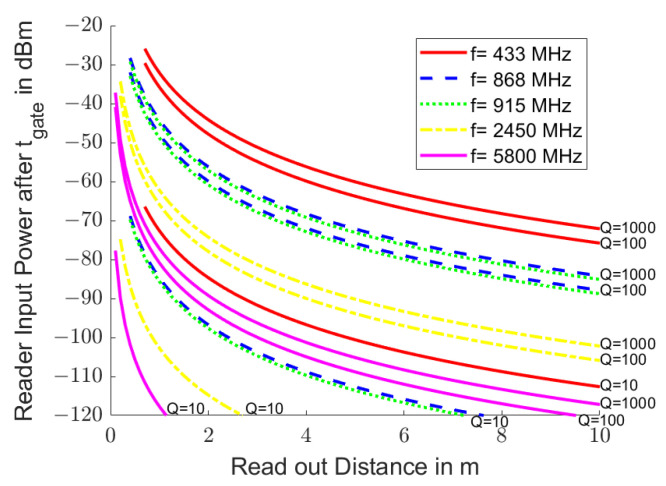
Remaining initial power level $P_{gated}$ in the reader when reading passive resonant transponders with a quality factor of 10, 100 and 1000 after a time gate to eliminate ambient echoes. The quality factor of the transmission channel was set to 15. There are three curves for each of the frequencies 433 MHz (red line), 868 MHz (green line), 915 MHz (blue line), 2.45 GHz (magenta line) and 5.8 GHz (black line) for the resonator quality factors of 1000 (solid line), 100 (dashed line) and 10 (dotted line). All other parameters correspond to those in Figure 3.

**Figure 6 sensors-25-00747-f006:**
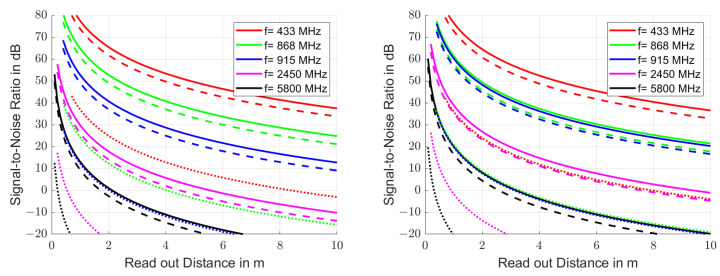
Remaining signal-to-noise ratio η in the reader after the time gate at the start of the sampling. The same frequency ranges and resonator qualities were used as in Figure 5. The noise bandwidths in the left graph were selected according to the regulations for the ISM [41] and SRD [42] bands, to 1.75 MHz for the 433 MHz band (red line), 2 MHz for the 868 MHz band (green line), 26 MHz for the 915 MHz band (blue line), 100 MHz for the 2.45 GHz band (magenta line) and 150 MHz for the 5.8 GHz band (black line). The resonator quality factors used were 1000 (solid line), 100 (dashed line) and 10 (dotted line). In the right graph, constant relative bandwidths of 0.5% of the carrier frequency are assumed. All other parameters correspond to those in Figure 3.

**Figure 7 sensors-25-00747-f007:**
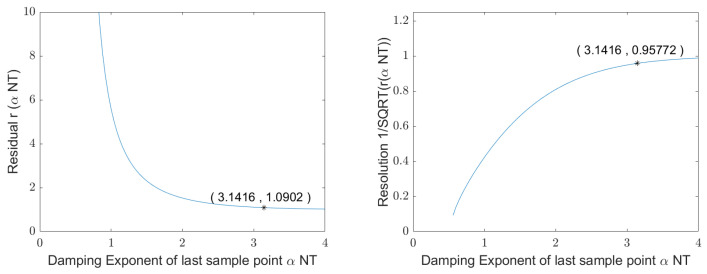
**Left chart**: reduction in the residual $r\left(\alpha NT\right)$ according to Equation (Equation 39), **right chart**: increase in the resulting measurement resolution, both graphs as a function of αNT. A resonator signal with a quality factor of 3 and a sampling of 2 samples per oscillation was used.

**Figure 8 sensors-25-00747-f008:**
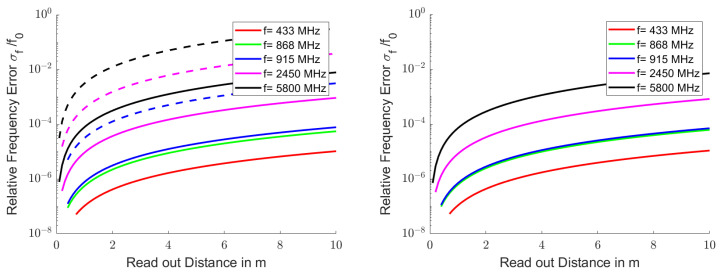
Crámer–Rao lower bound for the relative frequency error $\frac{\sigma_{f_{0}}}{f_{0}}$ as a function of the readout distance. The same frequency ranges, resonator quality factors and noise bandwidth as in Figure 6 were used in the left diagram, in the right graph, constant relative bandwidths of 0.5% of the carrier frequency are assumed. All other parameters correspond to those in Figure 3.

**Figure 9 sensors-25-00747-f009:**
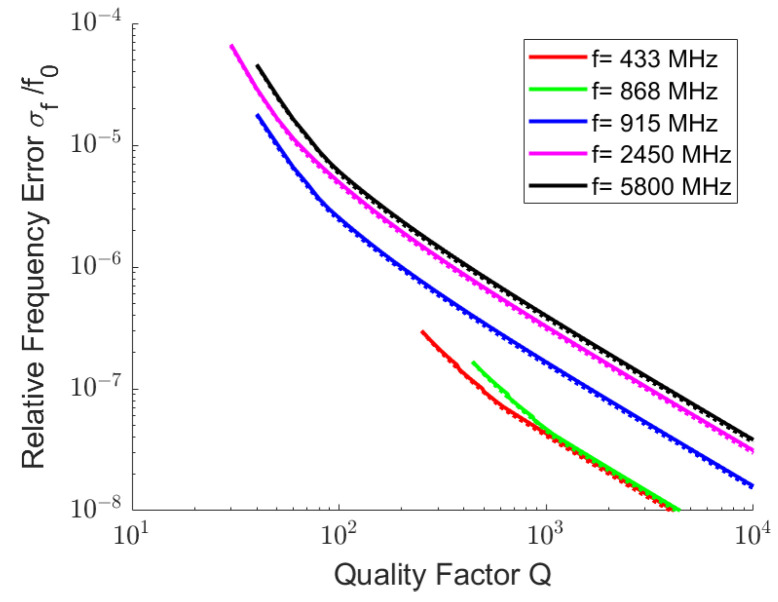
Relative frequency error $\sigma_{f_{0}}/f_{0}$ for one readout cycle as a function of the quality factor of a resonator in selected ISM or SRD bands. The read distance *R* is set to R=1λ. The diagrams start with the quality factor at which the resonance bandwidth fits into the assigned band. The resonator was stimulated and recorded for the time of *Q* oscillations. For quality factors where the resonator bandwidth became smaller than one fifth of the assigned bandwidth, the sampled bandwidth was limited to five times the resonator bandwidth. All other parameters correspond to those in Figure 3. The approximation in Equation (Equation 46) is also shown in the dotted lines.

**Figure 10 sensors-25-00747-f010:**
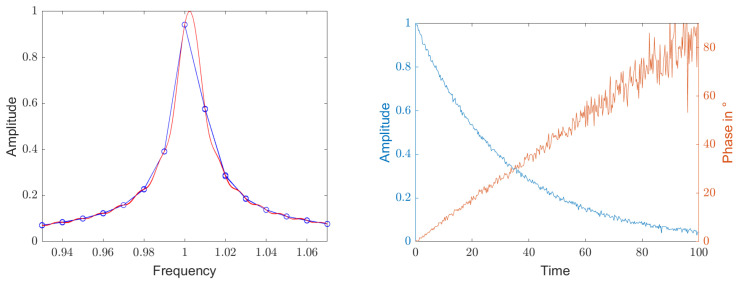
**Left**: Detail enlargement of the peak in the frequency domain that was calculated via Fourier transformation of the signal of a decaying resonator over Qf oscillations. The initial signal-to-noise ratio is 40 dB. The blue circles represent the sample values for a calculation without zero padding. The red line was calculated with a zero padding of 10. The center frequency is marked with a small cross. The slight ripple results from the truncation of the exponential function. **Right**: Measurement signal in magnitude and phase that was demodulated with the frequency of the coarse resolved maximum in the left spectrum without zero padding. Since the actual frequency is about a quarter step size apart from the maximum, a phase error of 90° remains.

**Figure 11 sensors-25-00747-f011:**
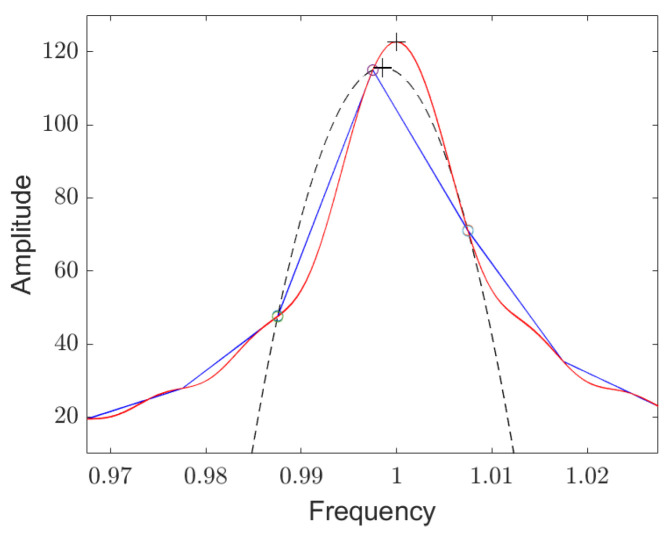
Detailed enlargement of the peak in the frequency domain, which was calculated by Fourier transforming the signal of a decaying resonator for Qf oscillations. The initial signal-to-noise ratio is 40 dB. The blue line connects the samples which were calculated without zero padding. The circles mark the maximum and its left and right sampling points. The dashed black line is a parabola approximation through these 3 points. The red line was calculated with a zero padding of 20 times the number of points. The actual resonance frequency and the maximum of the parabola are marked with small crosses.

**Figure 12 sensors-25-00747-f012:**
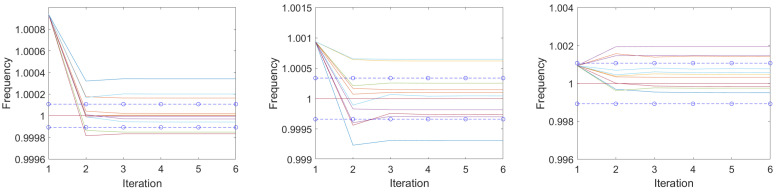
Convergence of above algorithm for 20 dB (**left**), 10 dB (**middle**) and 0 dB (**right**) signal-to-noise ratio according to Equation (Equation 19). The resonator has a center frequency of 1 and a quality of 100. ±1σ of the Crámer–Rao lower bound is shown in blue dashed lines. The estimated center frequencies start on the left at the nearest sampling point of the spectrum calculated via FFT and then converge very quickly to the best estimate for this data set. Each time, 10 examples with varied noise are shown. Note the different scaling of the ordinates of the graphs.

**Figure 13 sensors-25-00747-f013:**
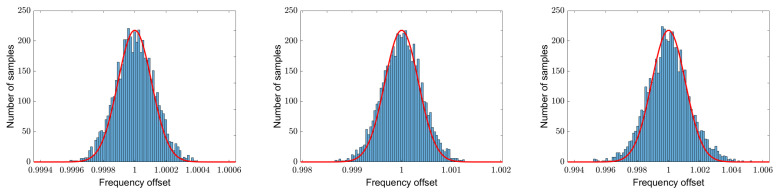
Histogram of 5000 runs of the simulations shown in Figure 12, plotted against ±6σ of the Crámer–Rao lower bound. The signal-to-noise ratios again are 20 dB (**left**), 10 dB (**middle**) and 0 dB (**right**). The normalized Gaussian functions of the corresponding Crámer–Rao lower bounds are also shown in red.

**Figure 14 sensors-25-00747-f014:**
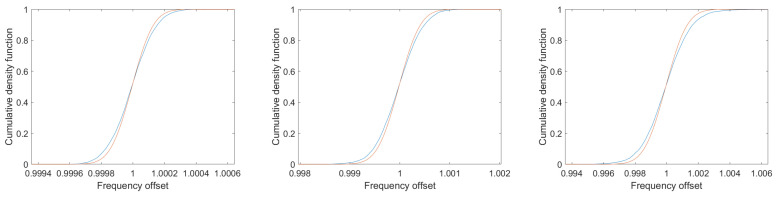
Cumulative density functions in blue of the 5000 simulation runs shown in Figure 13 together with the cumulative density functions of the corresponding Gaussians in red. The signal-to-noise ratios again are 20 dB (**left**), 10 dB (**middle**) and 0 dB (**right**).

**Figure 15 sensors-25-00747-f015:**
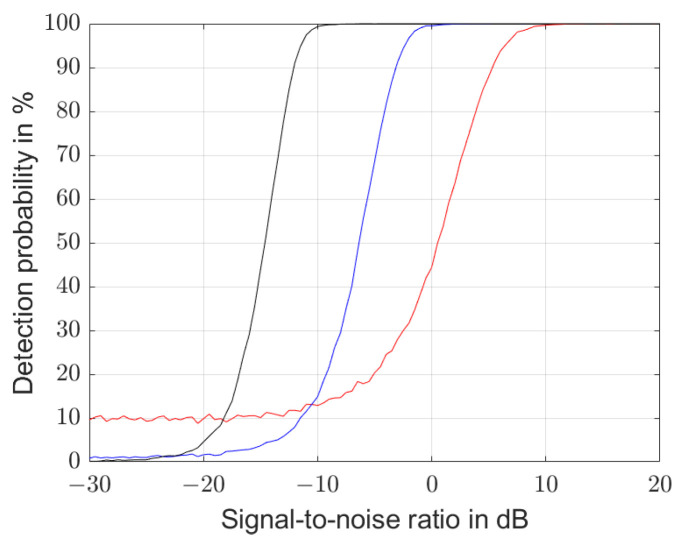
Probability that the algorithms in Equation (Equation 50) to Equation (Equation 53) result within an interval of ±6σ at the actual resonant frequency depicted as a function of the signal-to-noise ratio for a resonator with quality factors of 10 (red line), 100 (blue line) and 1000 (black line).

**Figure 16 sensors-25-00747-f016:**
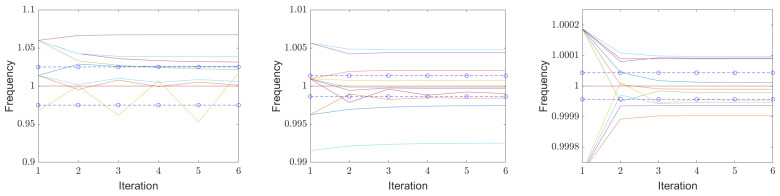
Convergence trajectories for each of the 10 runs at signal-to-noise powers where the detection probabilities are only 80% for a resonator with quality factors of 10 (**left**), 100 (**middle**) and 300 (**right**). ±1σ of the Crámer–Rao lower bound is again shown in blue dashed lines.

**Figure 17 sensors-25-00747-f017:**
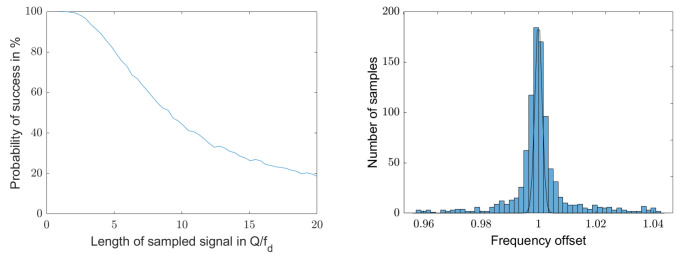
**Left**: Reduction in the convergence behavior of the algorithm as a function of the sample length for sample lengths between $1\cdot Q/f_{d}$ and $20\cdot Q/f_{d}$. The quality factor of the resonator signal is 100 and the initial signal-to-noise ratio is 0 dB. The probability of convergence of the algorithm decreases as the sample length increases from 100% to 20%. The **right** graph shows for the sample length of $20\cdot Q/f_{d}$ the few remaining correct detections of the natural frequency for 5000 runs.

**Figure 18 sensors-25-00747-f018:**
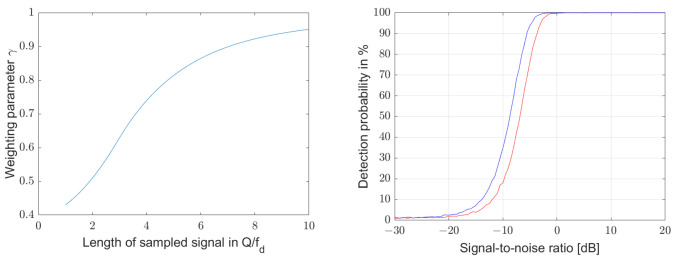
The left diagram shows the weighting coefficient γ according to Equation (Equation 56) as a function of αNT. The improvement in the convergence probability of the algorithm by the weighting function is shown in the right diagram for a resonator signal with a quality factor of 100 and an initial signal-to-noise ratio of 0 dB. The red line was calculated without weighting and the blue line with weighting. The detection probability shifts by approximately 3 dB to lower signal-to-noise ratios.

**Figure 19 sensors-25-00747-f019:**
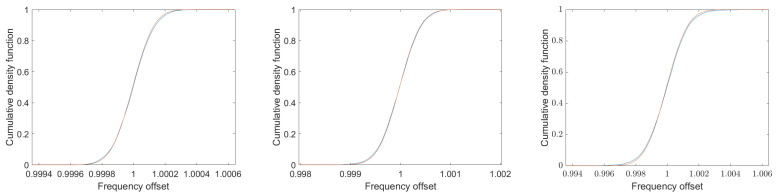
Cumulative density functions from 5000 simulation runs using the weighting function Equation (Equation 56) together with the cumulative density functions of the corresponding Gaussian functions. All other parameters correspond to those in Figure 14. The curves are so close together that they can hardly be distinguished. The signal-to-noise ratios are again 20 dB (**left**), 10 dB (**middle**) and 0 dB (**right**).

**Figure 20 sensors-25-00747-f020:**
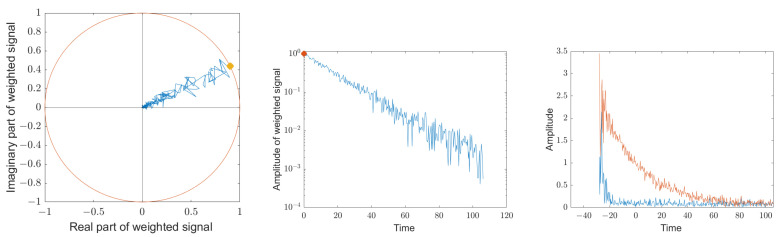
The left graph shows the remaining phase angles of the individual sampling points $z_{n}$, where the natural frequency was eliminated by multiplying with $e^{-j\tilde{\omega }nT}$ and where the impact of noise was reduced by weighting with $e^{-\alpha nT}$. In addition, the phase angle $\tilde{\varphi }$ at t=0 is plotted according to Equation (Equation 31). For the center graph, the amplitudes of the sample points weighted with $e^{-\alpha nT}$ were plotted together with the amplitude $\tilde{b}$ at t=0 calculated according to Equation (Equation 32). The noisy signal from the decaying resonator with a signal-to-noise ratio of 20 dB is shown in red on the right along with the echoes in the transmission channel. The blue curve shows the remaining signal when the estimated values $\mu_{n}$ have been subtracted from the measured values $z_{n}$.

**Figure 21 sensors-25-00747-f021:**
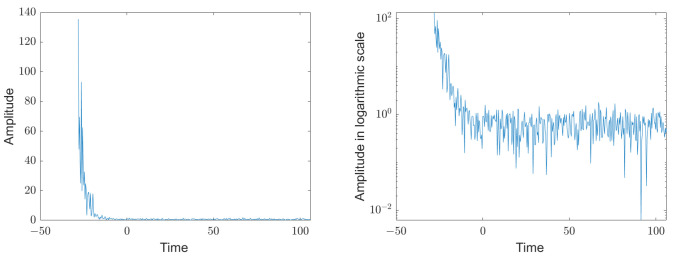
Sum of the complex values of 50 measurement signals where the estimated values of the decaying resonator have been subtracted; on the left in linear scale and on the right in logarithmic scale. The systematic interference signals disappear into the noise at t=−15, and the signal evaluation could be started earlier by this interval, resulting in a higher resolution.

## Data Availability

The data presented in this study and the corresponding MATLAB codes are available on request from the corresponding author.

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
