# Peer review of "Estimation of the Measurement Accuracy of Wireless Passive Resonance Sensors"

_sensors, 2025, doi:10.3390/s25030747_

Round 1
Reviewer 1 Report
Comments and Suggestions for Authors
- The authors conducted a study on the measurement accuracy of passive wireless resonant sensors. The obtained results are important, the article is well written but I have some remarks that need to be addressed to improve the work as a whole.
1. The abstract needs to explain a little about the method used (or proposed) for this study.
2. In Figure 2, what unit is time expressed in?
3. Same remark with figure 4.
4. Can you please provide more explanation regarding formula (16)?
5. Can you please discuss (in the conclusion) the limitations of your study?
6. There are some grammatical errors that appear in various sections of the manuscript.
- Addressing these points may improve the scientific quality of this study.
Comments on the Quality of English Language
- Overall, the quality of the English language is acceptable for this manuscript, however the authors can make more efforts to improve this quality.
Reviewer 2 Report
Comments and Suggestions for Authors
The authors have done an excellent job proposing a mathematical model that provides a theoretical estimate of the accuracy of a passive resonant sensor. However, I believe several comments could enhance the article.
1) Adding "estimate" to the title would provide more specificity.
2) In the introduction, the authors reference studies on estimating the accuracy of wireless sensors; however, they do not explain why a new model is needed. Additionally, the final section fails to present the results of comparing the new model and existing ones. If the new model is indeed unique, it would be beneficial to highlight its distinctive features in both the conclusion and the abstract.
3) The power graphs obtained at the same frequency but with different sensor Q-factors could be more effectively distinguished. Please include additional notations in Figures 5 and 6 to clearly indicate the Q values at which the graphs were obtained.
4) The comments in Figure 6 clarify the noise band values for different frequency ranges, but the source of these values remains unclear.
5) Line 341 states that for periodic signals with a constant amplitude, the variance approaches zero as the number of samples increases to infinity. When discussing signals with a finite spectrum, it appears that the Nyquist criterion is sufficient.
6) The manuscript does not describe the modeling processes, and the necessity of reference 34 is unclear.
7) It is advisable to incorporate an analysis of recent works related to the study topic over the past few years.
8) How can the incomplete matching of the sensor with the antenna affect the resulting model?
Comments on the Quality of English Language
The English could be improved to express the research more clearly. For example, there are two variants "readout" and "read-out". Please check the text for typos. Thank you!
Round 2
Reviewer 1 Report
Comments and Suggestions for Authors
- The remarks have been carefully addressed.
Comments on the Quality of English Language
- The quality of English Language is improved.
Author Response
The authors thank the reviewers for their valuable advice
Reviewer 2 Report
Comments and Suggestions for Authors
Thank you for answering all of the comments. However, there are still some problems with the quality of English. For example, the solid line is named "full line" in the caption for Figure 5.
Author Response

(The authors gave the same response as above.)
